# From Genetics to Clinical Implications: A Study of 675 Dutch Osteogenesis Imperfecta Patients

**DOI:** 10.3390/biom13020281

**Published:** 2023-02-02

**Authors:** Silvia Storoni, Sara J. E. Verdonk, Lidiia Zhytnik, Gerard Pals, Sanne Treurniet, Mariet W. Elting, Ralph J. B. Sakkers, Joost G. van den Aardweg, Elisabeth M. W. Eekhoff, Dimitra Micha

**Affiliations:** 1Department of Internal Medicine Section Endocrinology, Amsterdam UMC Location Vrije Universiteit Amsterdam, De Boelelaan 1117, 1081 HV Amsterdam, The Netherlands; 2Rare Bone Disease Center Amsterdam, De Boelelaan 1117, 1081 HV Amsterdam, The Netherlands; 3Department of Human Genetics, Amsterdam UMC Location Vrije Universiteit Amsterdam, De Boelelaan 1117, 1081 HV Amsterdam, The Netherlands; 4Department of Traumatology and Orthopeadics, University of Tartu, L. Puusepa 8, 50406 Tartu, Estonia; 5Department of Orthopedic Surgery, University Medical Center Utrecht, Heidelberglaan 100, 3584 CX Utrecht, The Netherlands; 6Department of Respiratory Medicine, Amsterdam University Medical Center, Location AMC, Meibergdreef 9, 1105 AZ Amsterdam, The Netherlands

**Keywords:** osteogenesis imperfecta, collagen type I, genetics, pathogenic variants, phenotype-genotype correlation, classification, hospital records

## Abstract

Osteogenesis imperfecta (OI) is a heritable connective tissue disorder that causes bone fragility due to pathogenic variants in genes responsible for the synthesis of type I collagen. Efforts to classify the high clinical variability in OI led to the Sillence classification. However, this classification only partially takes into account extraskeletal manifestations and the high genetic variability. Little is known about the relation between genetic variants and phenotype as of yet. The aim of the study was to create a clinically relevant genetic stratification of a cohort of 675 Dutch OI patients based on their pathogenic variant types and to provide an overview of their respective medical care demands. The clinical records of 675 OI patients were extracted from the Amsterdam UMC Genome Database and matched with the records from Statistics Netherlands (CBS). The patients were categorized based on their harbored pathogenic variant. The information on hospital admissions, outpatient clinic visits, medication, and diagnosis-treatment combinations (DTCs) was compared between the variant groups. OI patients in the Netherlands appear to have a higher number of DTCs, outpatient clinic visits, and hospital admissions when compared to the general Dutch population. Furthermore, medication usage seems higher in the OI cohort in comparison to the general population. The patients with a *COL1A1* or *COL1A2* dominant negative missense non-glycine substitution appear to have a lower health care need compared to the other groups, and even lower than patients with *COL1A1* or *COL1A2* haploinsufficiency. It would be useful to include the variant type in addition to the Sillence classification when categorizing a patient’s phenotype.

## 1. Introduction

Osteogenesis imperfecta (OI) is a rare hereditary connective tissue disorder characterized by bone fragility and skeletal deformities [1]. The disease, caused by monogenic mutations, leads to different levels of severity; therefore, using a classification system is inescapable [2]. The establishment of a genotype-based classification system that can accurately predict clinical features has been a challenge for the last several decades. The continuous discovery of new genetic causes for OI has resulted in a genotype-based classification system now listing more than 20 OI types [3,4]. The complexity of this system impedes its practicality, while a major issue remains in its inability to provide a description of distinct clinical types. OI is known to be characterized by great interfamilial and intrafamilial clinical variability, which means that it is currently not possible to safely attribute distinct clinical characteristics to each of the genetic OI types [4,5]. Consequently, the clinical classification system, known as the “Sillence classification,” is predominantly used worldwide [6]. The Sillence classification provides five OI types with clinical features ranging from mild to progressively deforming and lethal phenotypes [1,7,8,9]. Although the Sillence classification is currently the most convenient classification system, OI presentation within types still varies widely [9,10]. Furthermore, additional uncertainty is created by the fact that the clinical classification of patients can potentially change during the course of the disease and can be influenced by treatment [6,11,12].

OI is caused by alterations in the production of the collagen type I protein. Collagen type I is composed of two collagen α1(I) chains and one α2(I) chain [13], encoded by the *COL1A1* and *COL1A2* genes, respectively [14]. In approximately 85% of the OI cases, OI is caused by an autosomal dominant mutation in *COL1A1* or *COL1A2* [1]. The remaining cases are caused by mainly recessive variants in genes involved in the post-translational modification of the development of correctly folded collagen type I [1,9,15,16]. The phenotype of patients with variants in the *COL1A1* and *COL1A2* genes is known to be very diverse, possibly leading to all four of the five Sillence types [3]. Sillence type I is the mildest form of OI, and it is commonly caused by haploinsufficiency (HI) [17]. HI is the result of a nonsense, frameshift, or splice site pathogenic variant, causing a premature termination codon in the coding sequence; hence, a null-allele. With only one functional copy of the gene, fewer collagen α1(I) or α2(I) chains (depending on the affected gene) will be produced, resulting in a quantitative defect [18]. Sillence types II, III, and IV are thought to mostly be associated with a qualitative defect in collagen type I. Collagen type I is formed by the triple helix conformation of the two α1(I) and one α2(I) chains [15,17]. The formation of the helix is defined by a strict amino acid sequence pattern, in which glycine is present in every third residue [13,14]. Given its small side chain size, glycine can fit into the center of the collagen triple helix structure without helix distortion [19]. Studies have shown that clinical variations in severity are associated with the location of the variant as well as the affected chain and amino acid [1,10,20]. The structural abnormalities in collagen type I can result from different types of variants, for instance, missense variants, inframe deletions, and insertions. The missense substitutions can involve both glycine and other amino acids. When glycine is replaced, the formation and stability of the triple helix are damaged to a greater extent than compared to the substitution of other amino acids [21]. Consequently, missense substitutions of glycine are reported to occur more often in lethal cases of OI in comparison to other missense substitutions [17,21,22,23].

To date, a genotype-based classification system that can guide medical professionals with disease prognosis and treatment guidelines does not exist. It remains challenging to fully understand how pathogenic variants correlate with clinical features. Investigating the clinical consequences of different types of variants might give insight into the pathophysiological mechanisms underlying OI. In this study, clinical features, such as hospitalizations, outpatient clinic visits, and medication usage, are compared between patients with different pathogenic variant types. This study aims to give insight into the genotype-phenotype correlation, supporting the realization of a possible new classification system that combines both genetic and clinical features of OI. Additionally, this study provides a complete overview of health care demand in the Dutch OI population.

## 2. Methods

### 2.1. Study Participants

During the period between December 1991 and April 2021, all patients genetically diagnosed with OI at the national reference center for the molecular diagnosis of OI at the Amsterdam UMC were eligible for inclusion in this study. The pathogenic OI gene variants were identified in *COL1A1*, *COL1A2*, *CRTAP*, *TMEM38B*, *IFITM5*, *CREB3L1*, *FKBP10*, *PLOD2*, *SP7*, *SERPINF1*, *P3H1*, *BMP1*, and *PPIB*. [15] Pathogenic variants in *SERPINH1* and *KDELR2* were excluded since these variants had been found in the research setting and were not yet included in the diagnostic database. Based on the type of the pathogenic gene variant, patients were divided into six groups. The groups were haploinsufficiency (HI), dominant negative missense variants in an amino acid other than glycine in either the *COL1A1* or *COL1A2* gene (DN missense p.other), dominant negative missense variants of glycine substitution in the *COL1A1* gene (DN *COL1A1* missense glycine), dominant negative variants of glycine substitution in the *COL1A2* gene (DN *COL1A2* missense glycine), dominant negative inframe deletions or insertions in the *COL1A1* or *COL1A2* gene (DN inframe deletions or insertions) and recessive variants (recessive). These groups were determined based on the expected effects of the variants [9,17,24,25]. In addition to the patients’ genetic variant, the Sillence classification was provided as evaluated by the referring physician. The patients with OI type V (IFITM5) were excluded; due to the low patient number (<10 patients), this group could not be described.

### 2.2. Data Extraction

A cohort of 675 patients was extracted from the Amsterdam UMC Genome Database. CBS has anonymized health care data on the Dutch population, which is available for research upon request by health authorities and academic institutions. A match between the Amsterdam UMC Genome Database cohort and the CBS cohort was established in 95% (*n* = 644) of the OI patients. Information regarding patients’ age and, when applicable, age at time of death (available between 1995 and 2019) was extracted from the CBS database. In addition, data on “Diagnosis Treatment Combination” (DTC), outpatient clinic visits, number of X-rays, hospital admissions, and medication usage were extracted. The Dutch healthcare system makes use of DTCs. A DTC care product is the sum of all consultations, tests, and therapies a patient typically receives in a hospital for a specific diagnosis. Data on the registered medical specialty and the number of DTCs (available between 2013 and 2017) were retrieved. In addition, data on outpatient clinic visits and the number of X-rays were accessible from 2013 to 2017. Visits at the outpatient clinic included consultations and the provision of limited treatment, during which the patient was not admitted to the hospital. Information on the type of consultation (first or follow-up appointment, telephonic consultation, or emergency room visit) and registered medical specialty was extracted. Data on hospitalizations was available from 2013 to 2019. Information on the date of admission, type of admission (inpatient admission or day-care admission), the length of the admission, and the medical specialty at release were all collected. Data on medication usage was available in different years. This included information on the primary anatomical drug class to which the prescription medication belonged [26]. First-line medications, which might have been prescribed by both a general practitioner and a specialist, were included in the retrieved data; however, information on the duration of the prescription was not provided. As information on medication usage during different years could not be compiled due to possible (small) differences in which drugs were accepted as first-line medications, we chose to extract information on medication usage for the year 2017 only.

### 2.3. Data Analysis

The characteristics of the cohort (age and Sillence classification) were described for the total cohort as well as for the different genetic groups: haploinsufficiency, DN missense p.other, DN *COL1A1* missense glycine, DN *COL1A2* missense glycine, DN inframe deletions or insertions, recessive. The data is presented as absolute numbers and percentages. For age, the median and the 25th and 75th percentiles were reported.

In addition, the characteristics of hospital admissions, DTCs, and outpatient clinic visits are reported in absolute numbers and percentages. The yearly average number of hospital admissions, DTCs, outpatient clinic visits, and X-rays was reported for the cohort and for the different genetic groups using the mean, median, and standard deviation (SD). The average number of admissions specified per medical specialty per year was reported per 100 patient years. Generalized linear regression analyses were performed to estimate the effects of different pathogenic variants on the number of hospital admissions, number of DTCs, number of outpatient clinic visits, and number of X-rays (on average per year). As age could possibly serve as an effect modifier, when statistically significant, the patients’ mean age during follow-up was added to the model. *p*-values were adjusted for multiple testing using Tukey’s honest significance difference (HSD) test. A *p*-value of ≤0.05 was deemed statistically significant. The average number of drug prescriptions in 2017 was reported for the cohort as a whole as well as for seven age categories: 0 to 14, 15 to 24, 25 to 34, 35 to 44, 45 to 64, 65 to 74, and 75 years and older. The group of patients who were 75 years and older was analyzed but not described due to the low patient number. As group numbers would be too low, it was not possible to subdivide according to genetic group based on age.

When possible, health care data was compared to the general Dutch population using public data from the CBS (“www.cbs.nl (last accessed on 1 October 2022)”). Data on the number of hospital admissions, the number of DTCs, and the proportion of people using medication were available for the total Dutch population. Incidence rate ratios (IRR) were calculated for both hospital admissions and DTCs by genetic group for the OI population compared to the total Dutch population. The proportion of patients using medication in the OI cohort was compared to the proportion of people using medication in the total Dutch population based on the aforementioned age categories. The data is reported as percentages.

In order to ensure patient confidentiality, information regarding patient groups lower than 10 is not shown in the results. Groups with fewer than 10 patients are only shown when they cannot be directly traced back to single patients. Descriptive analyses were performed using IBM SPSS Statistics for Windows version 25 (IBM Corporation, Armonk, NY, USA), and for the generalized linear regression analysis, Rstudio v3.6.2 (RStudio: Integrated Development for R., PBC, Boston, MA, USA) was used.

### 2.4. Ethical Consideration

The Medical Ethics Review Committee (MERC) of the Amsterdam UMC (Amsterdam, The Netherlands) waived the need for ethics approval and the need to obtain consent for the analysis and publication of the retrospectively obtained and anonymized data for this non-intervention study (MERC study number 2021.0085).

## 3. Results

In total, the Amsterdam UMC genome database consisted of 675 patients genetically diagnosed with OI. Table 1 shows the different pathogenic variant types found in the cohort, including information on the affected gene, exon position, and general information about the type of variant. 353 patients (52%) had a HI variant, 45 patients (7%) had a DN missense p.other variant, 86 patients (13%) had a DN *COL1A1* missense glycine variant, 121 patients (18%) had a DN *COL1A2* missense glycine variant, 26 patients (4%) had a DN inframe deletion or insertion, and 35 patients (5%) had a recessive variant (Table 1). Information on the suspected Sillence classification was available for 73% (*n* = 490) of this cohort. Figure 1 shows the distribution of the gene variant groups and the Sillence classification.

A match between the Amsterdam UMC Genome Database cohort and the CBS cohort was established in 95% (*n* = 644) of the OI patients. In 2019, the median age of patients with a HI variant was 32 years [interquartile range (IQR): 13–48 years], patients with a DN missense p.other variant had a median age of 32 years [IQR: 19–50 years], patients with a DN *COL1A1* missense glycine had a median age of 31 years [IQR: 13–43 years], patients with a DN *COL1A2* missense glycine had a median age of 30 years [IQR: 9–44 years], patients with a DN inframe deletion or insertion had a median age of 33 years [IQR: 14–58 years], and patients with a recessive variant had a median age of 20 years [IQR: 5–47 years].

In total, 8587 DTCs were opened for OI patients between 2013 and 2017, of which 84% were opened during an outpatient clinic visit, 9% during an admission as a daycare patient, and 7% during an inpatient admission. In Table 2, the IRRs comparing the number of opened DTCs per person in the OI population to the number of DTCs opened in the total Dutch population are presented. For OI patients, a DTC was opened 2.6 times more often per person compared to the total Dutch population. Pairwise comparisons of the generalized linear regression analyses showed that patients with a HI variant had significantly fewer opened DTCs compared to patients with a DN inframe deletion or insertion (Tukey’s HSD adjusted *p*-value is 0.0078). In addition, statistically significant differences were found between the DN missense p.other and DN *COL1A1* missense glycine, DN inframe deletions or insertions, and recessive groups (Tukey’s HSD adjusted *p*-values are 0.0226, 0.0008, and 0.0220, respectively). In Table 3, the average number of DTCs per year for a patient of 30 years is reported for the different genetic groups.

During the period between 2013 and 2017, 14,315 outpatient clinic visits were registered (for 588 OI patients on average). A total of 26% were first-time outpatient clinic visits, 47% were follow-up outpatient clinic visits, and 9% were emergency clinic visits. OI patients had an average of 4.9 visits per year (median 3.8, SD 4.37). No statistically significant differences were found between the different genetic groups. Table 3 shows the average number of outpatient clinic visits reported per year for a 30-year-old patient. In addition, between 2013 and 2017, 4708 X-rays were performed. The mean number of X-rays per year was 1.60 (median 0.80, SD 2.11). Pairwise comparisons of the generalized linear regression analyses showed that patients with a DN missense p.other variant underwent statistically significantly fewer X-rays compared to patients with a DN *COL1A1* missense glycine, DN inframe deletions or insertions, and recessive variants (the Tukey’s HSD adjusted *p*-values are 0.0192, 0.0198, and 0.0340, respectively). Table 3 shows the average number of X-rays performed per year for a 30-year-old patient in each genetic group.

Additionally, between 2013 and 2019, 2133 hospital admissions were reported in the OI population. A total of 26% of the OI population was never admitted; 41% had one to three admissions, 16% had four to six admissions; 9% had seven to ten admissions, and 7% had more than 11 admissions. The percentage of patients having at least one admission, regarding the different genetic groups, was 56% for DN missense p.other, 73% for HI, 75% for DN *COL1A2* missense glycine, 80% for DN *COL1A1* missense glycine, 78% for recessive, and 80% for DN inframe deletions or insertions. Table 2 shows the average number of hospital admissions per year per 100 people in the OI population. Patients with a HI or DN missense p.other variant had statistically significantly fewer hospital admissions compared to patients with a DN inframe deletion or insertion (the Tukey’s HSD adjusted *p*-values are 0.0076 and 0.0052, respectively). In addition, the IRR, representing the mean yearly admissions per OI patient compared to a member of the Dutch population, is reported for the whole OI group as well as per genetic group.

In 2017, a total of 1734 medication prescriptions were registered for 605 OI patients. The average number of drugs used per OI patient 0-14 years old was 1.2 (median 1.0, SD 1.4), which was 2.5 for 15–24 year old patients (median 2.0, SD 2.3), 2.6 for patients 25–34 years old (median 2.0, SD 3.0), 2.9 for patients 35–44 years old (median 2.0, SD 3.0), and 3.7 and 6.0, respectively, for the groups of 55–64 and 65–74 years old (55–64 yr median 5.0, SD 5.0; 65–74 yr median 7.0, SD 3.5). Figure 2 shows the proportion of patients (expressed in percentages) that used medication in 2017 per age category.

## 4. Discussion

OI is a connective tissue disorder caused by defects in genes involved in the correct formation of collagen type I [27]. The patients with OI present puzzlingly great variability in phenotype [5,11]. A clinical classification system is predominantly used in clinical practice as a genetic classification system confidently predicting clinical features and prognosis is not available. The aim of this study was to provide more insight in the correlation between the type of genetic defect and phenotype. This was performed by examining the medical data in different genetic groups. In doing so, we also aimed to provide a complete overview of the medical care demand among Dutch OI patients.

Additionally, when comparing patient health care demands across the different pathogenic variant groups, there appears to be a recurring pattern. Patients with a DN missense p.other variant seem to have a lower demand for medical care compared to patients with other pathogenic variants. Statistically significant differences were discovered for the number of opened DTCs between patients with DN missense p.other variants and patients with either a DN *COL1A1* missense glycine variant, a DN inframe deletion or insertion, or a recessive variant (Table 3). Limited knowledge exists about the effects and clinical severity of DN missense p.other variants, especially in relation to other pathogenic variant types. Geneticists can disagree on the level of pathogenicity. OI patients with p.other, non-glycine missense substitutions are reported to present with variable severity, including mild OI [28,29]. It has been established that the mildest OI type (Sillence type I) is caused by both quantitative (HI) and qualitative variants [17,25]. This was also found in our cohort (Figure 1C). It is interesting to note that in patients with a DN missense p.other variant, a higher percentage of the mildest OI form was detected, compared to patients with other DN variants. The proportion of Sillence OI type I seemed similar in DN missense p.other and HI (Figure 1C); however, due to missing Sillence classification data, this is difficult to assess. Previous research on differences between quantitative and qualitative variants in young adults with OI type I reported no differences in phenotype apart from shorter stature [30]. Although not statistically significant, in our cohort, patients with a DN missense p.other variant seem to have on average fewer DTCs, fewer outpatient clinic visits, and fewer hospital admissions per person per year compared to HI (Table 3). As HI is considered to induce the mildest form of OI, our findings seem to contradict past literature [9,17,24]. It is important to take into account that certain qualitative variants are also likely to cause a mild form of OI, similarly to some glycine substitutions, and treatment options could work differently in people with different genetic backgrounds [31]. This is, for example, the case in the study of Orwell et al., in which teriparatide was hypothesized to be more effective in patients with a quantitative variant and thus more beneficial in patients with OI type I [32]. Future studies are expected to shed light on the clinical characteristics of OI patients with p.other non-glycine missense substitutions.

In our cohort, patients with DN *COL1A1* missense glycine variants, DN *COL1A2* missense glycine variants, DN inframe deletions or insertions, or recessive variants all seemed to have a higher medical care demand in comparison to patients with a DN missense p.other or an HI variant. Patients with a DN inframe deletion or insertion exhibited the highest medical care demand (Table 2) compared to other pathogenic variant types. Statistically significant differences were found for the number of opened DTCs and the number of hospital admissions between patients with DN inframe deletions or insertions and patients with either a DN missense p.other or an HI variant (Table 3). The effect of deletions and insertions is mostly unknown. In the studies by Maioli et al. and Lindahl et al., out of 587 patients, only three had a deletion; none were reported to have an insertion [17,30]. Of these three patients, two had Sillence type IV, while the Sillence type of the other patient was not reported. According to Byers et al., the majority of deletions they encountered resulted in the lethal OI type, which led them to hypothesize that the disease severity of a deletion likely depends on its location and size [9]. In our cohort, patients had both smaller (<10 exons) and larger deletions and insertions (>10 exons). Due to the small group sizes, we were not able to differentiate based on the size and/or location of the mutations. In our cohort, patients with a DN *COL1A1* missense glycine variant appeared to have a higher need for medical care than patients with a DN *COL1A2* missense glycine variant. This is in accordance with earlier research, which investigated glycine-to-serine substitutions, the most frequent amino acid substitution [30]. Lindahl et al. identified a more severe phenotype in COL1A1 substitutions, which included lower BMD and stature and a higher presence of dentinogenesis imperfecta. Missense glycine substitutions in *COL1A1* were also shown to be more likely to be lethal in the prenatal period than missense glycine variants in *COL1A2* [24]. A recent analysis of the entries in the international OI database also demonstrated a higher risk of a lethal phenotype for carriers of glycine substitutions in *COL1A1* than *COL1A2* [23]. Based on previous literature, recessive variants are believed to cause a relatively severe phenotype [33,34]. This is comparable with our findings: patients with a recessive pathogenic variant demonstrated the second highest need for medical care. We were unable to examine possible differences between recessive genes because of their low number, which necessitated their consideration as one group.

During 2013 and 2019, the Dutch OI population seemed to have, on average, a higher need for hospital care compared to the total Dutch population. Compared to the total Dutch population, patients with OI appeared to have 2.6 times more DTCs and 2.7 times more hospital admissions; in addition, a higher proportion of the OI population used medication (Figure 2). The results of our study support the findings of a Danish study describing the morbidity and mortality of OI patients [35,36]. In addition to a higher demand for medical care, they found a higher risk of death due to respiratory diseases, gastrointestinal diseases, trauma caused by bone fractures, and diseases of the nervous system. Consistent with their findings, our study showed that relatively more OI patients were prescribed medication compared to the total Dutch population for the following categories: alimentary tract and metabolism, respiratory system, cardiovascular system, and nervous system medication (Figure 2). Relatively higher drug use in the aforementioned categories supports the role of extraskeletal complications in terms of morbidity and mortality in OI patients. Although extraskeletal complications in OI are still under-investigated, there is growing awareness concerning the impact of extraskeletal features on OI patients’ health [36,37,38,39,40,41,42,43]. It is important to consider that the category of alimentary tract and metabolic medication also includes vitamin D and calcium. Therefore, the higher use of medication in this category can also be attributed to the use of vitamin D and calcium. Moreover, since the reason for the prescription and the length of drug use are not known, we can only speculate that extraskeletal complications are the cause of the relatively higher number of patients using a certain medication. More research is needed to better understand how extraskeletal complications impact OI patients’ health.

Although a very large cohort of patients with OI was examined, subdividing the patients according to their pathogenic variants resulted in relatively small group sizes. Therefore, it was not possible to differentiate between the location of missense substitutions or to distinguish between deletions and insertions and their sizes. Due to the small group size, all recessive variants are currently combined in one group. For the Sillence classification, we relied on the information communicated by numerous healthcare professionals across the Netherlands who provided this information when requesting the molecular diagnosis. Misclassification of patients cannot be ruled out, considering that classification according to Sillence can be difficult for medical professionals who are not familiar with OI. There are also numerous strengths. The cohort included a large number of OI patients with available genetic characterization. The Amsterdam Genome Database provides a comprehensive overview of the genetic background of the Dutch OI population, since this database is estimated to contain 85% of the Dutch OI population [44]. Using this OI patient cohort, we were able to match genetic groups to health care data. We believe that the health care data we used can provide a derivative of disease severity. The number of outpatient clinic visits and the number of hospital admissions give a clear indication of the consequences of disease on the patients’ quality of life. In addition, DTCs and data on medication use contribute to the understanding of the healthcare needs of OI patients, about which very few studies have been previously conducted [17,30].

In conclusion, our study clearly showed a trend in OI patients with a DN missense p.other variant having the lowest average need for health care, possibly indicating a milder phenotype than HI. In terms of health care demand, DN missense p.other is followed in ascending order by HI, DN *COL1A2* missense glycine, DN *COL1A1* missense glycine, recessive variants, and DN inframe deletions or insertions. Dutch OI patients seem to have a higher need for medical care compared to the total Dutch population. The increased use of medication in the OI cohort highlights the importance of better understanding the role of extraskeletal features in OI patients, such as cardiac, pulmonary, gastroenterological, and nervous system complications. Currently, when classifying a patient’s phenotype, it would be useful to add the variant type in addition to the Sillence classification. Future research on the effect of the type of variant on the clinical characteristics of OI patients is important in order to understand how we can eventually improve the prediction of the clinical outcome in order to facilitate an accurate prognosis, treatment, and follow-up.

## Figures and Tables

**Figure 1 biomolecules-13-00281-f001:**
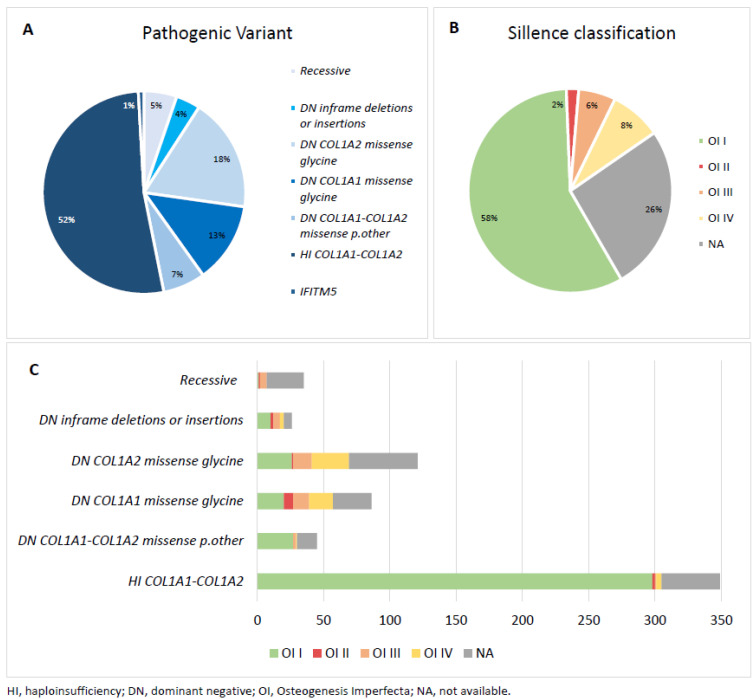
(**A**) The distribution of pathogenic variants found in the Dutch OI cohort. (**B**) The distribution of Sillence classification types in the Dutch OI cohort. (**C**) Combined overview of the Sillence classification type and pathogenic variants found in the Dutch OI cohort.

**Figure 2 biomolecules-13-00281-f002:**
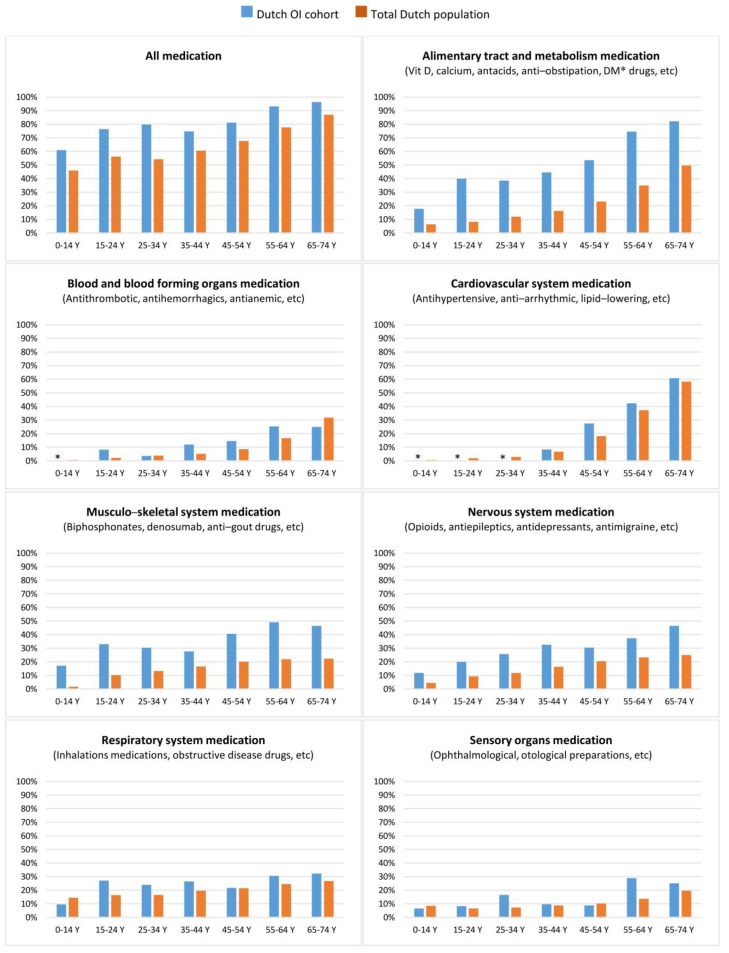
Proportion of the Dutch OI cohort and total Dutch population using medication belonging to specific ATC categories.OI, Osteogenesis Imperfecta; Y, year. * Due to low patient number exact percentage is not shown (* = <4%). Further information on ATC categories: “https://www.whocc.no/atc_ddd_index/” (last accessed on 14 November 2022).

**Table 1 biomolecules-13-00281-t001:** Pathogenic variants identified in the Dutch Osteogenesis Imperfecta cohort.

Pathogenic Variants			Total
			675
HI *COL1A1-COL1A2*			
	HI *COL1A1*		349
	HI *COL1A2*		<10
DN *COL1A1-COL1A2* missense p.other			45
	DN *COL1A1* missense p.other:	exon 1–5	<10
		exon 6–10	<10
		exon 11–42	<10
		exon 43–48	<10
		exon 49–51	<10
	DN *COL1A2* missense p.other:	exon 1–5	<10
		exon 6–10	<10
		exon 11–42	<10
		exon 43–48	<10
		exon 49–51	<10
DN *COL1A1* missense glycine			86
	DN *COL1A1* missense glycine:	exon 6–10	<10
		exon 11–42	63
		exon 43–48	<20
DN *COL1A2* missense glycine			121
	DN *COL1A2* missense glycine:	exon 1–5	<10
		exon 6–10	<10
		exon 11–42	108
		exon 43–48	<10
		exon 49–51	<10
DN inframe deletions or insertions			26
	DN *COL1A1* inframe deletion	exon 1–5	<10
		exon 6–10	<10
		exon 11–42	<10
	DN *COL1A1* inframe insertion:	exon 11–42	<10
	DN *COL1A2* inframe deletion:	exon 6–10	<10
		exon 11–42	<10
		exon 43–48	<10
	DN *COL1A2* inframe insertion:	exon 11–42	<10
		exon 43–48	<10
Recessive			35
	*CRTAP*, *TMEM38B*, *CREB3L1*, *FKBP10*, *PLOD2*, *SP7*, *SERPINF1*, *P3H1*, *BMP*, *PPIB*		
*IFITM5*			<10

HI, haploinsufficiency; DN, dominant negative. Due to patient confidentiality, group numbers lower than 10 cannot be shown.

**Table 2 biomolecules-13-00281-t002:** The number of diagnosis-treatment combinations and hospital admissions in the Dutch OI cohort compared to the total Dutch cohort.

A The Number of Diagnosis Treatment Combinations per Person in the OI Cohort Compared to the Total Dutch Population on Average during 2013–2017								
Medical Specialty	DN Missense p.Other	HI	DN *COL1A2* Missense Glycine	DN *COL1A1* Missense Glycine	Recessive	DN Inframe Deletions or Insertions	Total	
Total	1.8	2.4	2.8	3.1	3.3	4.1	2.6	
Ophthalmology	0.9	1.1	1.0	0.8	1.3	1.9	1.1	
Otolaryngology	1.7	2.3	2.5	1.4	1.3	3.3	2.2	
Surgery	1.9	3.2	2.3	3.4	1.7	2.0	2.9	
Surgery (plastics)	0.9	1.7	1.4	0.7	10.2	6.5	1.7	
Orthopedic surgery	4.6	6.3	8.2	9.3	9.2	13.2	7.4	
Urology	0.1	0.9	0.7	1.1	1.3	0.2	0.8	
Gynecology	2.1	1.5	2.3	1.4	0.6	1.0	1.6	
Dermatology	0.9	0.8	0.5	0.9	2.0	0.7	0.9	
Internal medicine	1.1	1.8	1.5	2.3	1.8	2.6	1.7	
Pediatrics	6.6	8.4	12.7	11.2	13.4	13.2	9.9	
Gastroenterology	1.3	0.8	0.6	0.5	0	0	0.7	
Cardiology	0.6	0.7	0.4	0.4	1.0	0.3	0.6	
Pulmonology	1.6	1.4	1.8	5.2	5.4	3.6	2.3	
Rheumatology	0.5	0.9	0.6	0.4	4.4	2.4	0.8	
Rehabilitation	11.5	15.3	25.4	29.2	28.0	54.0	20.9	
Geriatrics	0	2.0	0	1.4	1.3	0	1.2	
Anesthesiology	0.3	1.3	1.9	3.0	2.6	2.7	1.6	
Clinical genetics	17.6	22.8	31.2	24.4	34.6	45.8	25.8	
Audiology	3.7	7.5	8.7	9.0	0	13.9	7.5	
B The number of hospital admissions per 100 people on average per year in the dutch OI cohort during 2013–2019								
Medical specialty	DN missense p.other	HI	DN *COL1A2* missense glycine	DN *COL1A1* missense glycine	recessive	DN inframe deletions or insertions	Total	
Total	36.0	59.8	85.4	91.5	112.4	124.0	71.5	per 100 people
Pediatrics	12.0	20.9	41.6	42.2	61.4	49.6	29.6	per 100 people
Surgery	4.0	9.8	6.4	10.5	11.0	10.4	9.0	per 100 people
Orthopedic surgery	8.4	10.3	18.5	23.1	22.8	28.0	14.4	per 100 people
Internal medicine	0.9	2.5	4.3	3.4	1.4	24.8	3.7	per 100 people
Neurology	0	1.7	1.7	3.4	2.8	3.2	1.9	per 100 people
Otolaryngology	2.7	3.1	1.5	2.0	4.1	1.6	2.6	per 100 people
Ophthalmology	0	1.1	0.4	0.3	2.1	1.6	0.9	per 100 people
Gynecology	3.1	2.7	5.7	2.6	0.7	0.8	3.1	per 100 people
Gastroenterology	4.0	2.5	1.3	0.6	0.7	0.8	2.0	per 100 people
Cardiology	0.4	1.4	0.8	0.6	1.4	0	1.0	per 100 people
Pulmonology	0	1.4	0.8	0.9	0	0.8	1.0	per 100 people
Urology	0	0.9	0.8	0.9	0.7	0.8	0.8	per 100 people
Dentistry	0.4	0.6	0.9	0.9	2.8	0.8	0.8	per 100 people
Other	0	0.9	0.8	0.3	0.7	0.8	0.7	per 100 people
The number of hospital admissions per person in the OI cohort compared to the total dutch population on average between during 2013–2019								
Mean yearly admission rate in total Dutch population	0.19							
Mean yearly admission rate in the OI cohort	0.26 (median: 0.1, std. dev: 0.4)	0.43 (median: 0.3, std. dev: 0.5)	0.61 (median: 0.3, std. dev: 0.9)	0.65 (median: 0.3, std. dev: 0.8)	0.8 (median: 0.4, std. dev: 1.1)	0.89 (median: 0.6, std. dev: 1.0)	0.51 (median: 0.3, std. dev: 0.7)	
Incidence rate ratio OI cohort vs. total Dutch population	1.35	2.25	3.21	3.44	4.23	4.66	2.69	

A. Numbers should be interpreted as follows: an OI patient belonging to a specific genetic group has X times more chance of having a DTC in use compared to a person in the total Dutch population during 2013–2017. B. In the first part, the numbers should be interpreted as follows: 100 OI patients belonging to a specific genetic group had X hospital admissions on average during 2013–2019. In the second part, the number of hospital admissions in the Dutch OI cohort is compared to the number of hospital admissions in the total Dutch population. Numbers should be interpreted as follows: an OI patient belonging to a specific genetic group has X times more chance of having a hospital admission compared to a person in the total Dutch population during 2013–2019. DTC, diagnosis-treatment combination; OI, Osteogenesis Imperfecta; DN, dominant negative; HI, haploinsufficiency.

**Table 3 biomolecules-13-00281-t003:** The average number of units of medical care used per year for different genetic groups of the Dutch OI cohort.

Hospital Admissions (2013–2019)		Exp (Estimate)	95% CI	
	HI	0.41	0.33–0.50	ⱡ
	DN missense p.other	0.24	0.13–0.45	ꬸ
	DN *COL1A1* missense glycine	0.58	0.42–0.81	
	DN *COL1A2* missense glycine	0.53	0.40–0.71	
	DN inframe deletions or insertions	0.89	0.56–1.41	ⱡ,ꬸ
	Recessive	0.66	0.43–1.00	
Diagnosis treatment combinations (DTCs) (2013–2017)		Exp (estimate)	95% CI	
	HI	2.68	2.32–3.08	ⱡ
	DN missense p.other	2.04	1.52–2.72	ꬸ,ꬷ,†
	DN *COL1A1* missense glycine	3.41	2.76–4.21	ꬷ
	DN *COL1A2* missense glycine	3.06	2.54–3.67	
	DN inframe deletions or insertions	4.59	3.34–6.30	ⱡ,ꬸ
	Recessive	3.76	2.78–5.08	†
Outpatient clinic visits (2013–2017)		Exp (estimate)	95% CI	
	HI	4.66	4.01–5.41	
	DN missense p.other	3.73	2.79–4.99	
	DN *COL1A1* missense glycine	5.51	4.39–6.91	
	DN *COL1A2* missense glycine	4.80	3.95–5.83	
	DN inframe deletions or insertions	6.37	4.47–9.08	
	Recessive	5.74	4.13–7.98	
X-rays (2013–2017)		Exp (estimate)	95% CI	
	HI	1.37	1.13–1.67	
	DN missense p.other	0.91	0.60–1.40	ꬸ,ꬷ,†
	DN *COL1A1* missense glycine	1.93	1.44–2.59	ꬷ
	DN *COL1A2* missense glycine	1.66	1.29–2.14	
	DN inframe deletions or insertions	2.30	1.46–3.61	ꬸ
	Recessive	2.09	1.38–3.16	†

Presented is the data of the pairwise comparisons of the generalized linear regression analyses. The numbers should be interpreted as follows: a 30-year-old osteogenesis imperfecta patient belonging to a specific genetic group has on average X medical care units per year. Medical care units are the number of hospital admissions, the number of opened DTCs, the number of outpatient clinic visits, or the number of X-rays. ⱡ, ꬸ, ꬷ, † show if a statistically significant difference was found between two groups. DTC, diagnosis-treatment combination; OI, Osteogenesis Imperfecta; HI, haploinsufficiency, DN, dominant negative.

## Data Availability

The original contributions presented in the study are included in the article. Further inquiries can be directed to the corresponding author.

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
