# Peer review of "From Genetics to Clinical Implications: A Study of 675 Dutch Osteogenesis Imperfecta Patients"

_biomolecules, 2023, doi:10.3390/biom13020281_

Round 1

Reviewer 1 Report

The paper provides good data analysis on the OI patient in Dutch. I believe a similar analysis can be made for other patients in different other countries that will provide more insight into this type of disease. 

Well done. 

Author Response

Response:

We want to thank the Reviewer for his high appreciation of our manuscript and for his kind words. We agree that it would be interesting to have similar information about patients with osteogenesis imperfecta from other countries.

Reviewer 2 Report

The authors aim at establishing a genetics-based classification of Osteogenesis Imperfecta using clinical data from the OI population in Holland as well as the causing genetic variant. Their analysis makes a straight-forward paper with some interesting results (e.g. DN variants having a milder clinical outcome than null variants), a previously unavailable survey of the clinical needs of the OI population in the country, as well as providing the foundation for a re-classification of OI presentations where the genotype is included.

I think the paper can be accepted as is. I have only a few comments for the authors which I don't think are necessary to 'resolve' for the paper to be published:

- The authors use the background medication usage and other "clinical even ts" to evaluate the differences between age groups in OI patients. I wonder whether centre-based differences may be also relevant here. What level of variability is there between centres in, say, hospital admissions per 1000 inhabitants in its service area per year? Has it been ruled out that there is a bias due to inter-centres differences?

- I wonder whether the authors have considered some sort of self-classification method, something as simple as k-means clustering. If this is employed on the clinical data - will the patients be classified naturally into the genetic groups, or into the Sillence system, or neither? How good would be those systems to predict the clinical outcome, then?

- Is there whole-genome or -exome sequence data for OI patients? Part of the variability could be explained by polymorphisms in multiple other loci (which is a whole other endeavour, it is just a bit surprising that no approach using genomics has been mentioned).

Author Response

Response:

We thank the Reviewer for carefully evaluating our manuscript and for the very positive comments.

  1. The Reviewer is wondering whether center-based differences create bias, which can influence the outcome, for instance, regarding the number of hospital admissions. Although interesting, we did not compare data between different centers. In order to answer our research question, we focused on the demand for medical care regardless of the type of center the patient received care from. Furthermore, we believe that admission criteria between Dutch centers are comparable, so the chance of inter-center bias (which influences the research question) is low. In addition, due to privacy concerns, we are not able to determine the hospital center origin of patients, and thus, unfortunately, we have no control over this aspect.

  2. We would like to thank the Reviewer for his suggestion. In order to make the groups usable in clinical practice, we choose to create the groups based on the type of pathogenic variant. The comparison of their clinical characteristics took place after creating the groups. Establishing a methodology based on self-classification is an excellent point which we are keen to address in the future.

  3. Unfortunately, currently we do not have whole-genome or exome sequence data for our OI patients. The genetic analyses for the patients included in our database were performed due to the clinical suspicion of osteogenesis imperfecta. Therefore, also depending on the time of testing, either the SANGER method or the NGS panel (next-generation sequencing) was used. The NGS panel examines specifically genes involved in bone and connective tissue pathologies. We agree with the reviewer that it would be beneficial to have the whole genome sequenced in order to obtain more information on potential polymorphisms. We appreciate the feedback provided and we will consider incorporating whole genome sequencing in future studies.

Reviewer 3 Report

This manuscript analyzes the clinical and genetic variants of 675 patients with OI in the Netherlands in order to develop a more comprehensive new classification of OI disorders. The phenotype difference of OI patients with different gene variants was pointed out, and the correlation between genotype and phenotype was analyzed. In addition, this article summarizes the health-care needs of OI patients in the Netherlands by comparing information on hospital admissions, outpatient clinic visits, medication, and diagnosis-treatment combinations (DTC) with that of the general population.

1. In Abstract:

The number of words can be simplified, especially the method section. Besides, why two paragraphs used?

2. In Results:

1) Does the author use Excel for the figures? Their qualities are too low to be published in a fundamental research journal. Origin and other software need to use.   

2) Table 1: The repetitive parts of the table simplify the presentation and ensure clarity.

3) Figure 1: The picture is blurry, and the pie charts of Figures 1A and B can be adjusted and improved.

4) Table 2: Table preparation is not standard, and there are errors in punctuation marks of subheadings in Table 2A and B.

5) Three-line table style should be applied.

3. In Discussion:

1) The conclusion that the proportion of Sillence OI type I seemed similar in DN missense p.other and HI is mentioned, but the results are not shown.

2) More severe phenotypes of COL1A1 substitution need further discussed, including lower BMD and height and higher dentine hypoplasia. In addition, it is mentioned that the missense glycine variant in COL1A1 is more likely to be fatal in prenatal period than in COL1A2. However, these conclusions are not supported by data.

3) It is noted that more patients with OI receive prescription treatment for digestive tract and metabolic, respiratory, cardiovascular and neurological drugs than the total population of the Netherlands. But the data in this article do not fully show this.

Author Response

Response:

We thank the Reviewer for carefully evaluating our manuscript and the suggestions that have been instrumental in helping us improve the quality of our work. The Reviewer points out that the English language usage could be improved. We addressed this feedback by having the manuscript reviewed by a native English speaker. We are confident that the manuscript is now of a high standard and that any language issues have been addressed.

  1. Abstract:
    We thank the Reviewer for the comment regarding the abstract. We have simplified and shortened the methods section, and we have created a single paragraph.
  2. Results:
    1),3) We thank the Reviewer for the suggestion regarding the resolution of the images in our manuscript. We have made the necessary improvements. The images have been resized to a higher resolution, and new images have been added to the manuscript.
    2) We have improved the presentation of Table 1 as suggested by the Reviewer. We are confident that this suggestion ensures clarity.
    4),5) We have improved and corrected the punctuation errors in tables 2A and B as suggested by the Reviewer. We have changed the tables to a three-line style as suggested by the Reviewer.
  3. Discussion:
    1) We thank the reviewer for drawing to our attention the omission in the sentence mentioned. We have added the reference (Figure 1C) in line 36 of the discussion.
    2) We agree with the suggestion of the Reviewer. The Reviewer has noted that the phenotype of COL1A1 could be better argued and that some data in line 70 of the discussion are not supported by the results. The Reviewer is correct; these two sentences do not refer to the results of our study but rather to the results of the study that is cited in line 69. We have therefore revised and improved the text by adding the reference to improve the sentence.
    3) The Reviewer has pointed out that the use of medication for the digestive, metabolic, respiratory, cardiovascular, and neurological systems was not illustrated in the results. This information is presented in Figure 2. In this figure, we have described the quantity of medication used by patients with OI and by people in the general Dutch population. We have added a reference (Figure 2) to line 91 in the discussion section.

We thank the Editor for the opportunity to respond to the thoughtful comments of the reviewers. We have incorporated the suggestions into the manuscript.

Round 2

Reviewer 3 Report

The Figure quality can be improved using Origin or other software.

Author Response

Response:

We thank the Reviewer for carefully evaluating our manuscript